# The Antimutagenic and Antioxidant Activity of Fermented Milk Supplemented with *Cudrania tricuspidata* Powder

**DOI:** 10.3390/foods9121762

**Published:** 2020-11-28

**Authors:** Sae-Byuk Lee, Banda Cosmas, Heui-Dong Park

**Affiliations:** 1School of Food Science and Biotechnology, Kyungpook National University, 80 Daehakro, Daegu 41566, Korea; lsbyuck@nate.com (S.-B.L.); barack9016@gmail.com (B.C.); 2Institute of Fermentation Biotechnology, Kyungpook National University, 80 Daehakro, Daegu 41566, Korea

**Keywords:** fermented milk, yogurt, *Cudrania tricuspidata*, antimutagenic activity, antioxidant activity, probiotics

## Abstract

In this study, *Cudrania tricuspidata* (CT) containing abundant phytochemicals, such as xanthones and flavonoids, was evaluated as an additive to fortify the functionality and organoleptic quality of fermented milk. The physicochemical, functional, and sensory properties of fermented milk supplemented with different concentrations of CT powder were investigated. Increasing amounts of CT powder elevated the malic acid concentration, increasing the total acidity and decreasing the pH of fermented milk supplemented with CT powder. The viable cell count and free sugar contents of fermented milk indicated that supplementing with CT powder improved lactic acid fermentation slightly. The color of fermented milk supplemented with CT powder was darker, redder, yellower, and more pleasing than the control fermented milk. The total phenolic and flavonoid contents of fermented milk supplemented with CT powder rose as the concentration of supplemented CT powder increased, resulting in enhanced antioxidant and antimutagenic activities. The CT powder improved the functionality of the fermented milk; still, at 2% or more, it had some unfavorable sensory properties, such as sourness, taste, and texture, which reduced the overall consumer preference. Therefore, a CT powder concentration of 0.5% or 1% may be acceptable to consumers.

## 1. Introduction

Fermented milk, one of the popular functional foods, is typically produced by thermophilic lactic acid bacteria (LAB), such as *Streptococcus thermophilus* and *Lactobacillus delbrueckii* subsp. *bulgaricus*; abundant and live LAB are still present in the final fermented milk product [1]. The LAB are probiotics that balance between beneficial and undesirable microorganisms in the gastrointestinal tract, affect the immune system, and enhance health [2]. Fermented milk is an excellent source of nutrients, including protein, calcium, phosphorus, riboflavin, thiamin, vitamin B12, folate, niacin, magnesium, zinc, and bioactive peptides [1,3,4]. Several studies have revealed that bioactive peptides have good potential for preventing cancer [5,6,7], lowering blood pressure [8,9], and managing type 2 diabetes [10,11]. Adding fruits, vegetables, or herbs into fermented milk has been recently considered a great a strategy to satisfy varying consumer preferences by boosting the functionality and organoleptic quality of fermented milk [12,13,14,15,16].

*Cudrania tricuspidata*, a member of the *Moraceae* family mainly cultivated in China and Korea [17], can be a good additive for fermented milk. Several studies have reported that two major groups of phytochemicals, such as xanthones and flavonoids, are present in all parts of *C. tricuspidata*. Prenylated xanthones and flavonoids, primarily present in leaves and root bark, showed remarkable anti-inflammatory [18,19], antitumor [20,21], hepatoprotective [22,23], neuroprotective [24,25], and anticoagulant [26] activities. Hydroxybenzyl flavonoid glycosides, abundant in stem bark, were reported to be a natural antioxidant and potential antitumor agents [27,28]. Prenylated isoflavonoids and benzylated flavonoids in the fruits have anti-inflammatory and antioxidant activities [29,30]. Despite its numerous health benefits, *C. tricuspidata*‘s utilization has been focused on its leaf and root since its raw fruit has a distinctive flavor that can be unacceptable to consumers [31]. Furthermore, the fruit contains relatively less polyphenols and flavonoids than the other parts [32,33]. However, several recent studies indicate that adding an appropriate amount of *C. tricuspidata* can improve the organoleptic properties of various food product with significantly enhanced functional properties [31,34,35].

A mutagen is a chemical or physical agent that is responsible for a genetically critical damage in the DNA sequence of an organism [36]. Since a mutation is related to the initiation and progression phase of carcinogenesis, the development of an antimutagenic agent can be a promising way to prevent genetic disease in humans, including cancer [37]. Several studies have reported that various polyphenols and flavonoids in plants induce antimutagenic activity [38,39]. Moreover, a study by de Oliveria et al. revealed that xanthones and flavones of *Syngonanthus* could serve as a novel antimutagenic agents [40]. Since these compounds are also plentiful in *C. tricuspidata*, we speculated that *C. tricuspidata* could be a potential antimutagenic agent.

This study aims to study how *C. tricuspidata* improves the functional properties of fermented milk and optimize the amount of *C. tricuspidata* to be organoleptically acceptable to potential consumers. This study also examines the antioxidant and antimutagenic activities of fermented milk supplemented with different concentrations of *C. tricuspidata* powder. Moreover, *C. tricuspidata*’s physicochemical and sensory properties were evaluated.

## 2. Materials and Methods

### 2.1. Preparation of Cudrania Tricuspidata Powder

Fully ripe *Cudrania tricuspidata* fruits (15.2 ± 0.1 °Brix, pH 5.3 ± 0.1, acidity 0.20% ± 0.03%) were obtained from Sangju, South Korea, during the 2017 harvest season (Figure 1). *C. tricuspidata* fruits were selected from the uniformity of color and lack of decay or rot. The proximate compositions of *C. tricuspidata* fruit were analyzed by the means of Association of Official Analytical Chemists (AOAC) [41] (Table 1). Samples were washed five times with tap water to remove any black spot on the skin and then manually squeezed to obtain the juice. The juice was filtered with cheesecloth and centrifuged at 10,000× *g* for 10 min. The supernatant was collected and freeze-dried into powder.

### 2.2. Preparation of Fermented Milk Supplemented with Cudrania tricuspidata

First, the bulk starter was prepared using milk with 32 g/L protein, 32 g/L fat, and 1.05 g/L calcium (Seoul Dairy Co., Seoul, Korea). The milk was heat-treated at 95 °C for 15 min and cooled down to 30 °C. After that, the milk was inoculated with 0.3% (*w*/*w*) of a commercial yogurt starter powder containing *Streptococcus thermophilus*, *Lactobacillus paracasei*, and *L. delbrueckii* subsp. *bulgaricus* (Plain morning, Cellbiotech Co. Ltd., Gimpo, Korea). The mixture was incubated at 40 °C for 8 h and then stored at 4 °C.

Four percent (*w*/*w*) skim milk was added to milk to create the fermentation substrate for yogurt production. The mixture was heat-treated at 95 °C for 15 min and cooled down to 30 °C. Subsequently, *C. tricuspidata* powder was added to the mixture at the concentration of 0%, 0.5%, 1%, 2%, or 3% (*w*/*w*) (Table 2). The mixtures were mixed well and inoculated with 3% (*w*/*w*) of bulk starter and incubated at 40 °C for 12 h.

### 2.3. Assessment of Physicochemical Properties and LAB Viable Counts in the Fermented Milk

#### 2.3.1. Soluble Solid Contents, pH, Total Acidity, and Viable LAB Counts

All the fermented milk samples were prepared by centrifugation at 10,000× *g* for 10 min. Soluble solid contents were determined using a refractometer (RA250, ATAGO, Tokyo, Japan). The pH of fermented milk samples was measured with a pH meter (Mettler-Toledo, Schwerzenbach, Switzerland). Total acidity, expressed as % of lactic acid, was determined by titration with 0.1 N NaOH until neutralization to pH 8.2. Each sample was serially diluted with distilled water using an appropriate dilution factor to achieve around 30–300 CFU/plate to determine the viable cell count in fermented milk. Each sample was then spread onto MRS (Difco Laboratories, Detroit, MI, USA) agar media plates and incubated at 37 °C for 48 h.

#### 2.3.2. Total Phenolic and Flavonoid Compounds

The total phenolic compounds in fermented milk samples were determined with the Folin-Ciocalteau method [42], with some modifications. Two milliliters of samples were mixed with 2 mL of 50% (*v*/*v*) Folin-Ciocalteu reagent and incubated at room temperature for 3 min. Each mixture was then mixed with 2 mL of 10% Na_2_CO_3_, vortexed, and allowed to stand at room temperature for 1 h. The absorbance was measured at 700 nm. The results were expressed as the equivalent of gallic acid mg/mL in fermented milk using a standard curve with gallic acid (0–0.5 mg/mL). The total flavonoid content in fermented milk samples was determined [43]. The fermented milk samples were examined spectrophotometrically at 510 nm against a blank solution containing all reagents in 200 μL of distilled water instead of fermented milk samples (UV-1601, Shimadzu Co., Kyoto, Japan). First, 430 μL of 50% ethanol, 70 μL of fermented milk sample, and 50 μL of 5% NaNO_2_ were combined in a test tube. After 30 min of incubation, the samples were mixed with 50 μL of 10% Al(NO_3_)_3_·9H_2_O. Six minutes later, 500 μL of NaOH (1 N) was added, and the solutions vortexed. The results were expressed as the equivalents of quercetin mg/mL of fermented milk using a standard curve with quercetin (0–200 μg/mL).

#### 2.3.3. Free Sugar and Organic Acid Contents

Next, free sugar and organic acid contents of fermented milk samples were examined. Fermented milk samples were centrifuged at 10,000× *g* for 15 min, and the supernatant was filtered (Millex-HV 0.45 μm, Millipore Co., Bedford, MA, USA). Filtered samples were determined by high-performance liquid chromatography (HPLC) (Model Prominence, Shimadzu, Kyoto, Japan) with a Sugar-Pak I column (diameter 6.5 × 300 mm; Waters, Milford, MA, USA) and PL Hi-Plex H column (diameter 7.7 × 300 mm; Agilent Technologies, Santa Clara, CA, USA) [44]. The chromatography for the free sugars was run at a flow rate of 0.5 mL/min at 90 °C in the mobile phase in 50 mg/L Ca-ethylenediaminetetraacetic acid (Ca-EDTA) buffer. The chromatography for the organic acids was run at a flow rate of 0.6 mL/min at 65 °C in the mobile phase in 0.005 M sulfuric acid. Free sugars and organic acids were detected with a refractive index detector (RID-10A, Shimadzu, Tokyo, Japan). The results were expressed as each compound’s equivalents in mg/mL of fermented milk using a standard curve with each compound (0–10 mg/mL).

#### 2.3.4. Hunter’s Color Value

Changes in Hunter’s color value of the fermented milk before and after fermentation were measured with a colorimeter (Konica Minolta CM-3600A, Osaka, Japan) calibrated with a standard calibration slide (50 × 12 mm). The results were expressed as *L* (brightness), *a* (redness), and *b* (yellowness). The *L*, *a*, and *b* values for standard tile were (*L* = 97.78, *a* = 0.39, *b* = 2.05).

### 2.4. Assessment of Antioxidant Activities in Fermented Milk

All antioxidant activities were analyzed according to Oszmianski et al. [45]. For diphenylpicrylhydrazyl (DPPH) radical scavenging activity analysis, 100 μL of DPPH was dissolved in pure ethanol (96%). The radical stock solution was prepared right before experimentation. Then, 1 mL of DPPH was added to 1 mL of fermented milk and 3 mL of 96% ethanol. The mixture was thoroughly shaken and placed at room temperature in the dark for 10 min. The resulting solution’s change in absorbance was observed at 517 nm using a spectrophotometer (UV-1601, Shimadzu Co., Kyoto, Japan). The results were corrected for dilution and expressed in μM of Trolox/mL of fermented milk using a standard curve with Trolox (0–50 μM/mL).

2,2′-azino-bis(3-ethylbenzothiazoline-6-sulfonic acid (ABTS) was dissolved in water to make a 7-μM stock solution for the ABTS radical scavenging activity analysis. The ABTS stock solution reacted with a final concentration of 2.45 μM potassium persulfate to produce ABTS radical cations (ABTS^+^). The reaction was kept in the dark at room temperature for 12–16 h before use; the radical would remain stable in this condition for more than 2 days. The ABTS^+^-containing samples were diluted with double distilled water to an absorbance of 0.700 ± 0.02 at 734 nm and equilibrated at 30 °C. After adding 3.0 mL of diluted ABTS^+^ solution (A_734 nm_ = 0.700 ± 0.02) to 30 μL of fermented milk sample, the absorbance was read exactly at 6 min after the initial mixing (UV-1601, Shimadzu Co., Kyoto, Japan). The results were corrected for dilution and expressed in μM Trolox/1 mL of fermented milk using a standard curve with Trolox (0–50 μM/mL).

The ferric ion reducing antioxidant power (FRAP) assay was based on an antioxidant’s tendency to reduce. An antioxidant reduces ferric ions (Fe^3+^) to ferrous ions (Fe^2+^) to form a blue complex (Fe^2+^/TPTZ) with increased absorbance at 593 nm. Moreover, the FRAP reagent was prepared by mixing 300 μM acetate buffer at pH 3.6, 10 μM of TPTZ in 40 μM of HCl, and 20 μM of FeCl_3_ at a ratio of 10:1:1 (*v*/*v*/*v*). An amount of 300 μL of FRAP reagent and 10 μL of fermented milk samples were mixed in each well. The absorbance was measured at 593 nm after 10 min (UV-1601, Shimadzu Co., Kyoto, Japan). A standard curve was plotted using different Trolox concentrations. All solutions were prepared on the same day of experimentation. The results were corrected for dilution and expressed in μM of Trolox/1 mL of fermented milk using a standard curve with Trolox (0–50 μM/mL).

### 2.5. Assessment of the Antimutagenic Activity of Fermented Milk

The antimutagenic activity of fermented milk supplemented with *C. tricuspidata* powder was investigated using the pre-incubation method [44]. Two mutagens, N-methyl-N’-nitro-N-nitrosoguanidine (MNNG) and 4-nitro-O-phenylenediamine (NPD) (Sigma Co., St. Louis, MO, USA), were dissolved in dimethyl sulfoxide (DMSO) to the final concentrations of 5 and 15 μg/plate, respectively. *Salmonella typhimurium* histidine operon mutant strains derived from LT-2, TA98 (hisD3052 rfa ΔuvrB) (TA98), and TA100 (hisG46 rfa ΔuvrB) (TA100), were also used. One hundred microliters of a fermented milk sample, 100 μL of MNNG or NPD solution, 100 μL of an overnight culture of TA98 or TA100, and 0.5 mL of 0.2 M sodium phosphate buffer (pH 7) were mixed in a capped glass tube. The mixture was then incubated at 37 °C for 30 min with agitation. Then, 3 mL of molten agar containing histidine and biotin was added to the mixture; the final mixtures were plated on a minimal glucose agar medium and incubated at 37 °C for 48 h in the dark. Subsequently, the number of His^+^ revertants per plate was counted. The antimutagenic activity was expressed as follows:Antimutagenic inhibition ratio (%) = (A − B)/(A − C)(1)

A was the number of mutagen-induced His^+^ revertants in the absence of a sample, B was the number of mutagen-induced His^+^ revertants in the presence of a sample, and C was the number of spontaneous His+ revertants.

### 2.6. Sensory Evaluation

Sensory evaluation of fermented milk samples was performed by a panel of twenty judges with sensitive taste discrimination from the Department of Food Science and Technology, Kyungpook National University, Korea. Each judge evaluated the color, odor, sourness, taste, texture, and overall preference of the fermented milk samples with intervals of at least 3 min between samples. Water was provided to cleanse the judges’ palates. Sensory scores were assigned on a scale ranging from 1 (dislike extremely) to 7 (like extremely). Before sensory evaluation was conducted, the experimental plan was approved by Kyungpook National University Industry Foundation on 1 October 2018 (IRB approval number: 2018-0160). All panels were given explanation and agreed with the precaution related to the possible side effects of *C. tricuspidata* before performing the sensory evaluation.

### 2.7. Statistical Analysis

All experiments were conducted in triplicate or more; the data were analyzed using the SAS software (version 9.4, SAS Institute, Cary, NC, USA). Significance was determined at a threshold of *p* < 0.05 using ANOVA, followed by Duncan’s multiple range test.

## 3. Results and Discussion

### 3.1. Physicochemical Properties and Viable Cell Count of Fermented Milk

The various physicochemical properties, such as soluble solids content, pH, total acidity, and viable cell count, of fermented milk supplemented with different concentrations of *C. tricuspidata* powder were investigated (Table 3). The soluble solids content of all the fermented milk samples was 9.8 °Brix. The pH decreased, and total acidity increased in the fermented milk samples as the concentration of supplemented *C. tricuspidata* powder increased since various organic acids of *C. tricuspidata* were concentrated by freeze-drying. The viable cell counts of all fermented milk samples were slightly increased depending on the supplemented amount of *C. tricuspidata* powder, indicating some components in *C. tricuspidata* powder contribute minimally to promoting lactic acid fermentation. Total phenolic compounds and total flavonoid contents of fermented milk supplemented with *C. tricuspidata* increased from 3.7- to 12.8-fold and from 10.1- to 30.10-fold when compared with the control fermented milk, respectively. *C. tricuspidata* fruit has been reported to contain chlorogenic acid, rutin, and other phenolic compounds as well as prenylated and benzylated forms of flavonoids [29,46,47]. In addition, five parishin derivatives (gastrodin, parishin A, B, C, and E) were newly identified and detected at high amounts in *C. tricuspidata* fruit [48]. Many plant phytochemicals, including polyphenols and flavonoids, have been recently defined as prebiotics since they meet the criteria, and many studies have demonstrated that health benefits are more attributed to the metabolites produced by microorganisms than the parent polyphenolic compounds [49,50]. It has been reported that supplementing various fruit juices (aronia, blueberry, and grape) into yogurt enhances the viability of *S. thermophilus*, indicating that increased phenolics may act as prebiotics [51]. Nguyen and Hwang also reported that viable count of LAB in yogurt increased during lactic acid fermentation with increasing supplemented aronia juice [1]. *C. tricuspidata* and *Morus alba L.* leaf extracts have also been reported to shorten fermentation time and increase the viability of LAB, which should be related to the phenolic compounds present in the extracts [35]. In the present study, *C. tricuspidata* fruit powder containing high amounts of phenolics and flavonoids may increase the growth of LAB in fermented milk.

Glucose and fructose have been reported to be the main components of *C. tricuspidata* [52]. In free sugar analysis, the glucose and fructose contents of the fermented milk samples rose with the increasing concentration of supplemented *C. tricuspidata* powder, except for 0.5% *C. tricuspidata* powder. LAB typically hydrolyzes lactose to glucose and galactose; however, several LAB, such as *S. thermorphilus* and *L. delbrueckii* subsp. *bulgaricus*, are galactose negative (Gal^-^) strains and can metabolize only glucose, thereby accumulating galactose in the milk [53]. This is because these strains do not have Tagtose-6P pathway and do not metabolize galactose [54]. The lactose content of fermented milk decreased, while galactose content increased as the concentration of the *C. tricuspidata* supplement increased. This data supported that *C. tricuspidata* powder affected lactic acid fermentation as viable cell count of fermented milk supplemented with *C. tricuspidata* powder increased. In the organic acid analysis, malic acid contents of the fermented milk were significantly increased as the concentration of *C. tricuspidata* supplement increased. Malic acid has been reported as one of the primary organic acids in *C. tricuspidata* [52] and the cause of sour and astringent tastes in food [55]. Lactic acid contents of fermented milk showed a slightly increasing trend, depending on the *C. tricuspidata* supplement concentration, which is due to lactic acid fermentation promoted by the addition of *C. tricuspidata* powder. The increases in malic and lactic acid contents caused a decrease in pH values and increased total acidities of fermented milk supplemented with *C. tricuspidata* powder.

### 3.2. Measuring Color Changes in Fermented Milk Supplemented with Cudrania tricuspidata

Color is one of the most critical parameters for consumer preference. Although *C. tricuspidata*’s fruit offers less potential health benefits than its leaf or root bark, it can be an excellent additive for various food products due to its pleasant coloring. The color values of fermented milk supplemented with different concentrations of *C. tricuspidata* powder before and after fermentation were investigated (Table 4). The *L* values, the brightness of the product, significantly decreased with increasing concentration of *C. tricuspidata* powder. Moreover, *L* values in all the fermented milk samples were slightly reduced after fermentation since pH decreased during fermentation. According to García-Pérez et al., lowering pH decreased the lightness in fermented milk due to gelation [56]. The *a* value, the indicator of redness, significantly increased with an increasing amount of *C. tricuspidata* powder. Additionally, the *a* values in most fermented milk samples, except for 0.5% of *C. tricuspidata* powder, were slightly increased after fermentation. Several studies have described that *a* values of yoghurt increased when various fruits such as grape, aronia, blueberry, and cherry, which contain a large amount of anthocyanin, were added [51,57]. Kim et al. also reported that the total carotenoid content of *C. tricuspidata* fruit increased during ripening, leading to the significantly higher *a* value compared to *C. tricuspidata* fruit in the lower maturity stages [48]. The *b* values, the yellowness indicator (if *b* > 0), increased significantly with an increasing amount of *C. tricuspidata* powder. Likewise, *b* values were higher in all the fermented milk samples after fermentation. Novruzov and Agamirov [58] reported that *C. tricuspidata* fruit contains various carotenoids, leading to an increase in *b* values of fermented milk supplemented with *C. tricuspidata* powder. Overall, the results of Hunter’s color value indicated that the fermented milk supplemented with *C. tricuspidata* powder were redder, darker, and more yellow than the control fermented milk as the concentration of *C. tricuspidata* increased. Along with Hunter’s color value, appearances of fermented milk supplemented with *C. tricuspidata* powder were also pleasant (Figure 2).

### 3.3. Functional Properties of Fermented Milk Supplemented with Cudrania tricuspidata

The DPPH radical scavenging, ABTS radical scavenging, and FRAP activities of fermented milk samples were analyzed to determine the effect of *C. tricuspidata* supplement on the functional properties of fermented milk (Table 5). With the *C. tricuspidata* supplement, the DPPH radical scavenging activity increased from 0.63 to 1.46 μM TE/mL, the ABTS scavenging activity increased from 0.18 to 0.36 μM TE/mL, and the FRAP activity increased from 0.21 to 1.65 μM TE/mL compared to the control fermented milk. In other words, the antioxidant activities of fermented milk supplemented with *C. tricuspidata* increased in a concentration-dependent manner. *C. tricuspidata* leaf extract has been considered an excellent source for high-value-added food materials and functional foods [16,59,60,61]; the leaves have a high level of polyphenols, especially quercetin, which have higher antioxidant activity than other parts of the plant [62,63]. On the other hand, *C. tricuspidata* fruit’s antioxidant activity is strongly affected by maturation and associated with the changes of prenylflavonoids, such as artocarpesin, alpinumisoflavone, 6-isopentenylgenistein, 4′-*O*-methylalpinumisoflavone, and 6,8-diprenylgenistein; fully matured *C. tricuspidata* fruits reach its peak of antioxidant activity [64]. The present study revealed that the addition of *C. tricuspidata* powder improved the antioxidant activity of fermented milk in vitro, indicating that *C. tricuspidata* has promising potential as a functional food. In vivo testing should also be performed to verify its functionality.

The antimutagenic activity of fermented milk supplemented with different concentrations of *C. tricuspidata* powder was investigated (Table 6). MNNG and NPD were used as mutagens for *Salmonella typhimurium* TA 100 strain. The antimutagenic activities, or the inhibition rate, of fermented milk supplemented with different amounts of *C. tricuspidata* powder increased from 18.75% to 64.58% for MNNG and from 23.31% to 60.11% for NPD. As with *S. typhimurium* TA 98 strain, the inhibition rate of fermented milk supplemented with different amounts of *C. tricuspidata* powder increased from 9.01% to 54.05% for MNNG from 20.00% to 63.81% for NPD. Like antioxidant activity, antimutagenic activities of fermented milk supplemented with *C. tricuspidata* were also concentration-dependent. Fermented milk prepared with LAB has been known to be antimutagenic toward a broad spectrum of mutagens such as MNNG, 4-nitroquinoline N-oxide (NQNO), and 3,2-dimethyl-4-aminobiophenyl (DMAB) [65,66]. Milk proteins, such as casein, R-lactalbumin, and β-lactoglobulin, have been shown to bind mutagenic heterocyclic amines at a high percentage [67,68]. Another study reported a robust inhibitory activity by casein against NQNO, and casein’s anti-NQNO activity increased when it was hydrolyzed by pepsin [69]. Matar reported that antimutagenic compounds are produced in milk during fermentation by *L. helveticus*, and the author suggested that the release of peptides is one possible contributing mechanism [70]. On the other hand, several studies have focused on antimutagenic activity of some phytochemical compounds, such as flavonoids (e.g., quercetin) and phenolic compounds (e.g., tannin) which are ubiquitous in plants [71,72]. The present study also identified *C. tricuspidata*’s antimutagenic activity due to its abundant phenolic compounds and flavonoids; therefore, *C. tricuspidata* could be an antimutagenic functional food.

### 3.4. Sensory Evaluation

The sensory aspects of fermented milk supplemented with different concentrations of *C. tricuspidata* powder were evaluated to determine the acceptable concentration of *C. tricuspidata* for consumer preference (Table 7). The color scores of fermented milk became higher as the concentration of supplemented *C. tricuspidata* powder increased, possibly due to the various pigments, such as carotenoids and anthocyanins, in *C. tricuspidata*. The odor scores of all the fermented milk supplemented with *C. tricuspidata* were higher than the control. However, the sourness, taste, and texture scores of fermented milk with the *C. tricuspidata* supplement exhibited different patterns. The fermented milk sample supplemented with 0.5% *C. tricuspidata* powder obtained the highest score in sourness and taste; however, higher concentrations of *C. tricuspidata* negatively impacted these scores due to the increase in malic acid concentration (Table 3). The supplementation *C. tricuspidata* powder also lowered the texture scores; only the fermented milk supplemented with 0.5% of *C. tricuspidata* powder was similar to the control. In overall preference, fermented milk supplemented with 0.5% of *C. tricuspidata* powder obtained the highest score. Of all the sensory properties, except for texture, the fermented milk supplemented with 1% of *C. tricuspidata* powder was not significantly different from the control even though it obtained lower sourness and taste scores. Therefore, the fermented milk supplemented with 0.5% of *C. tricuspidata* powder had the highest consumer preference. Moreover, fermented milk supplemented with 1% of *C. tricuspidata* powder can also be considered as a potential functional food for the health-conscious consumers since it contains 1.5–3 times higher antioxidant activities and 2–3 times higher antimutagenic activities than the control fermented milk.

## 4. Conclusions

In the present study, we evaluated various physicochemical and functional properties of fermented milk supplemented with *C. tricuspidata* and found that bioactive compounds, such as total phenolic and flavonoid contents of fermented milk, were increased by the addition of *C. tricuspidata* powder. Moreover, sensory evaluation was performed to determine the appropriate concentration of supplemented *C. tricuspidata* powder for optimal consumer preference. Various antioxidant activities, including DPPH radical scavenging, ABTS radical scavenging, and FRAP activities, and antimutagenic activity, were significantly enhanced as the concentration of supplemented *C. tricuspidata* powder increased. In sensory evaluation, color scores of fermented milk samples were raised with the increase in *C. tricuspidata* concentration. However, most sensory parameter scores such as sourness, taste, texture, and overall preference dramatically declined as the concentration of *C. tricuspidata* reached or exceeded 2%. Therefore, supplementing *C. tricuspidata* powder at 0.5% or 1% may be more acceptable to consumers while still providing good functional properties. Further in vivo study will prove the enhanced functionalities of fermented milk supplemented with *C. tricuspidata* powder.

## Figures and Tables

**Figure 1 foods-09-01762-f001:**
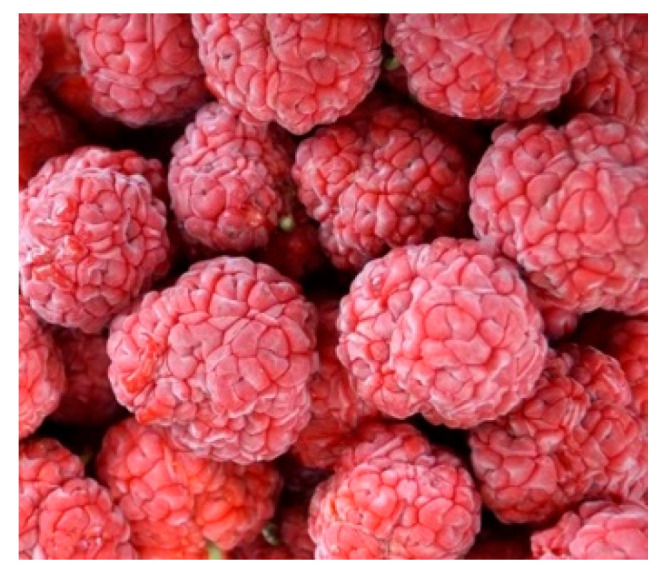
Image of *Cudrania tricuspidata* used in this study.

**Figure 2 foods-09-01762-f002:**
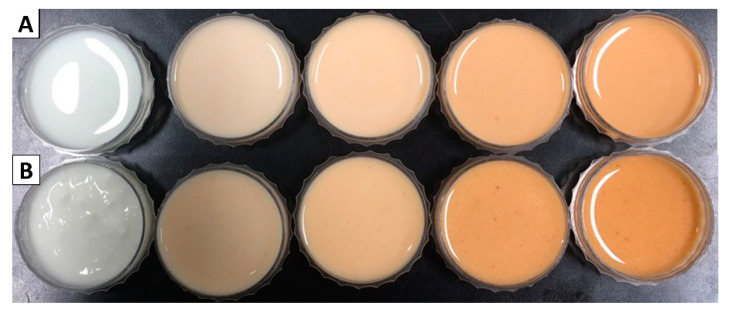
Images of fermented milk supplemented with different concentrations of *Cudrania tricuspidata* powder before (**A**) and after fermentation (**B**). From left to right, the fermented milk supplemented with 0%, 0.5%, 1%, 2%, or 3% of *C. tricuspidata* powder.

**Table 1 foods-09-01762-t001:** The proximate compositions of *Cudrania tricuspidata* fruit.

Proximate Composition (%)	
Moisture	80.16 ± 1.43
Carbohydrate	12.40 ± 0.43
Crude protein	2.40 ± 0.32
Crude lipid	1.50 ± 0.04
Crude ash	3.54 ± 0.44

All data are expressed as the mean ± Standard Deviation (SD) (*n* = 3).

**Table 2 foods-09-01762-t002:** The mixing ratios of fermented milk supplemented with *Cudrania tricuspidata* powder.

Ingredient (g)	*Cudrania tricuspidata* Powder Content (%, *w*/*w*)				
	0	0.5	1	2	3
Raw milk	930	925	920	910	900
Skim milk	40	40	40	40	40
Bulk starter	30	30	30	30	30
*Cudrania tricuspidata* power	0	5	10	20	30

**Table 3 foods-09-01762-t003:** Physicochemical properties and viable LAB (lactic acid bacteria) cell count of fermented milk supplemented with *Cudrania tricuspidata* powder.

Parameter	*Cudrania tricuspidata* Powder Content (%, *w*/*w*)				
	0	0.5	1	2	3
Soluble solid (°Brix)	9.8 ± 0.0 ^a^	9.8 ± 0.0 ^a^	9.8 ± 0.1 ^a^	9.8 ± 0.0 ^a^	9.8 ± 0.0 ^a^
pH	4.51 ± 0.02 ^a^	4.47 ± 0.01 ^b^	4.40 ± 0.02 ^b^	4.34 ± 0.01 ^d^	4.30 ± 0.01 ^e^
Total acidity (%)	1.11 ± 0.02 ^b^	1.13 ± 0.02 ^b^	1.19 ± 0.01 ^b^	1.23 ± 0.03 ^ab^	1.28 ± 0.02 ^a^
Viable LAB cell count (Log CFU/mL)	9.01 ± 0.11 ^a^	9.05 ± 0.25 ^a^	9.08 ± 0.19 ^a^	9.16 ± 0.23 ^a^	9.21 ± 0.18 ^a^
Total phenolic compound (mg GE/mL)	1.77 ± 0.04 ^e^	6.62 ± 0.07 ^d^	8.09 ± 0.29 ^b^	16.50 ± 0.59 ^b^	22.57 ± 1.03 ^a^
Total flavonoid content (μg QE/mL)	5.55 ± 2.52 ^e^	55.81 ± 8.58 ^d^	80.26 ± 8.24 ^b^	126.13 ± 1.26 ^b^	167.05 ± 8.73 ^a^
Free sugar contents (mg/mL)					
Glucose	ND	ND	3.66 ± 0.24 ^c^	6.21 ± 0.19 ^b^	8.90 ± 0.31 ^a^
Fructose	ND	ND	1.13 ± 0.08 ^c^	4.20 ± 0.13 ^b^	6.94 ± 0.25 ^a^
Lactose	54.59 ± 2.31 ^a^	51.57 ± 2.18 ^ab^	48.34 ± 1.96 ^b^	47.75 ± 2.05 ^b^	47.00 ± 1.83 ^b^
Galactose	4.93 ± 0.24 ^b^	4.84 ± 0.31 ^b^	5.49 ± 0.34 ^ab^	6.07 ± 0.41 ^a^	5.86 ± 0.29 ^a^
Organic acid contents (mg/mL)					
Malic acid	7.47 ± 0.37 ^c^	6.54 ± 0.56 ^c^	7.40 ± 0.38 ^c^	11.43 ± 0.52 ^b^	14.96 ± 0.43 ^a^
Lactic acid	11.85 ± 0.21 ^b^	11.75 ± 0.28 ^b^	11.92 ± 0.20 ^b^	12.32 ± 0.22 ^ab^	12.54 ± 0.16 ^a^

^a–e^ Different letters within the same column indicate a significant difference (*p* < 0.05). All data are expressed as the mean ± SD (*n* = 3). ND, not detected.

**Table 4 foods-09-01762-t004:** Hunter’s color values of fermented milk supplemented with *Cudrania tricuspidata* powder before and after fermentation.

Color Value	Fermentation	*Cudrania tricuspidata* Powder Content (%, *w*/*w*)				
		0	0.5	1	2	3
*L*	Before	78.63 ± 0.13 ^a^	66.73 ± 0.99 ^b^	63.29 ± 0.24 ^c^	58.14 ± 0.08 ^d^	54.24 ± 0.06 ^e^
	After	75.06 ± 0.99 ^a^	64.61 ± 1.54 ^b^	61.27 ± 0.33 ^c^	56.58 ± 0.60 ^d^	53.24 ± 0.04 ^e^
*a*	Before	−2.40 ± 0.03 ^e^	6.50 ± 0.17 ^d^	9.97 ± 0.11 ^c^	13.09 ± 0.13 ^b^	14.77 ± 0.03 ^a^
	After	−1.72 ± 0.07 ^e^	6.40 ± 0.30 ^d^	10.40 ± 0.19 ^c^	15.91 ± 0.28 ^b^	17.77 ± 0.04 ^a^
*b*	Before	5.55 ± 0.25 ^e^	13.76 ± 0.25 ^d^	17.56 ± 0.25 ^c^	21.09 ± 0.23 ^b^	22.03 ± 0.01 ^a^
	After	7.24 ± 0.40 ^e^	16.14 ± 0.37 ^d^	21.13 ± 0.21 ^c^	26.62 ± 0.33 ^b^	28.08 ± 0.09 ^a^

^a–e^ Different letter within the same column indicate significant difference (*p* < 0.05). All data are expressed as the mean ± SD (*n* = 3).

**Table 5 foods-09-01762-t005:** Antioxidant activities of fermented milk supplemented with *Cudrania tricuspidata.*

Antioxidant Activity (μM TE/mL)	*Cudrania tricuspidata* Powder Content (%, *w*/*w*)				
	0	0.5	1	2	3
DPPH radical scavenging activity	1.94 ± 0.12 ^c^	2.57 ± 0.20 ^b^	2.77 ± 0.14 ^b^	3.18 ± 0.10 ^a^	3.40 ± 0.11 ^a^
ABTS scavenging activity	0.31 ± 0.02 ^c^	0.49 ± 0.03 ^b^	0.52 ± 0.05 ^ab^	0.64 ± 0.08 ^a^	0.67 ± 0.04 ^a^
FRAP activity	0.19 ± 0.01 ^e^	0.40 ± 0.02 ^d^	0.66 ± 0.04 ^c^	1.20 ± 0.08 ^b^	1.84 ± 0.13 ^a^

^a-e^ Different letter within the same column indicate significant difference (*p* < 0.05). All data are expressed as the mean ± SD (*n* = 3).

**Table 6 foods-09-01762-t006:** Antimutagenic activities (inhibition rate, %) of fermented milk supplemented with *Cudrania tricuspidata* powder against N-methyl-N’-nitro-N-nitrosoguanidine (MNNG) and 4-nitro-O-phenylenediamine (NPD) on *Salmonella* typhimurium TA100 and TA98.

Strain	Mutagen	*Cudrania tricuspidata* Powder Content (%, *w*/*w*)				
		0	0.5	1	2	3
TA100	MNNG	21.23 ± 1.45 ^e^	39.98 ± 0.99 ^d^	57.88 ± 0.56 ^c^	70.17 ± 3.69 ^b^	85.81 ± 1.30 ^a^
	NPD	17.42 ± 1.34 ^e^	40.73 ± 3.15 ^d^	52.53 ± 3.46 ^c^	66.57 ± 2.04 ^b^	77.53 ± 2.25 ^a^
TA98	MNNG	16.52 ± 2.32 ^d^	25.53 ± 4.45 ^c^	49.55 ± 1.78 ^b^	64.56 ± 2.62 ^a^	70.57 ± 3.55 ^a^
	NPD	24.76 ± 2.50 ^e^	44.76 ± 2.13 ^d^	56.19 ± 2.24 ^c^	74.29 ± 1.52 ^b^	88.57 ± 3.58 ^a^

^a–e^ Different letter within the same column indicate a significant difference (*p* < 0.05). All data are expressed as the mean ± SD (*n* = 3).

**Table 7 foods-09-01762-t007:** Sensory evaluation of fermented milk supplemented with *Cudrania tricuspidata* powder.

Sensory Property	*Cudrania tricuspidata* Powder Content (%, *w*/*w*)				
	0	0.5	1	2	3
Color	3.60 ± 1.28 ^b^	4.15 ± 1.24 ^b^	4.90 ± 1.70 ^ab^	5.50 ± 1.71 ^a^	5.40 ± 1.86 ^a^
Odor	4.30 ± 1.23 ^b^	4.80 ± 0.91 ^ab^	4.60 ± 1.12 ^ab^	4.70 ± 1.29 ^ab^	5.20 ± 1.33 ^a^
Sourness	4.40 ± 1.62 ^a^	4.60 ± 1.42 ^a^	3.75 ± 1.25 ^ab^	3.10 ± 1.25 ^bc^	2.35 ± 1.04 ^c^
Taste	4.30 ± 1.25 ^a^	4.45 ± 1.22 ^a^	3.95 ± 1.22 ^ab^	3.50 ± 1.24 ^b^	3.05 ± 1.28 ^b^
Texture	5.35 ± 1.41 ^a^	4.85 ± 1.31 ^a^	3.65 ± 1.19 ^b^	2.65 ± 1.14 ^c^	2.15 ± 1.01 ^c^
Overall preference	4.15 ± 1.26 ^a^	4.30 ± 1.05 ^a^	3.55 ± 1.12 ^ab^	3.10 ± 1.25 ^b^	2.70 ± 1.09 ^b^

^a–c^ Different letters within the same column indicate a significant difference (*p* < 0.05). All data are expressed as the mean ± SD (*n* = 3).

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
