# Peer review of "The Antimutagenic and Antioxidant Activity of Fermented Milk Supplemented with Cudrania tricuspidata Powder"

_foods, 2020, doi:10.3390/foods9121762_

Round 1

Reviewer 1 Report

The manuscript submitted by Lee et al. deals with the evaluation of antimutagenic and antioxidant activity of fermented milk supplemented with a powder of Cudrania tricuspidata, a plant containing abundant phytochemicals, such as xanthones and flavonoids.

The study is quite original and the experimental design is clear.

Major criticisms:

  • The English style must be revised.
  • for each evaluation, the number of samples analyzed and the experimental replicates performed must be clearly indicated.
  • The C. tricuspidata powder has not been characterized prior to use. The evaluations were made only downstream on fermented milk.
  • the discussion needs to be improved. Very often the authors limit themselves to saying only if their results are in agreement or not with what has already been reported in the literature.

Minor concerns:

L49: indicate which polyphenols and which flavonoids.

L50: are these compounds present in equal amount in all parts of the plant (leaves, fruits, roots, ...)?

L55: must be entered additional information on what is meant by mutagenic activity.

L61: remove “the”.

L62: “…were obtained from a local from in Sangju…” local company?

L71: Do you know the relative percentage of Streptococcus thermophilus, Lactobacillus paracasei and L. bulgaricus in the starter mixture?

L72: It is not correct to start a sentence with a number.

L75: write “C. tricuspidata” in italic.

L79: paragraph 2.3 must be divided into distinct sections: from L80 to L87, from L88 to L100, from 101 to L110, from L111 to L114.

L92-93: What is the range of the standard curve with gallic acid?

L99: What is the range of the standard curve with quercetin?

L115: remove “the”.

L122: What is the range of the standard curve with trolox?

L131: “Trolox/1 mL”. Remove “1” and indicate the range of the standard curve with trolox.

L141: “Trolox/1 mL”. Remove “1” and indicate the range of the standard curve with trolox.

L142: remove “the”.

L164-165: for each parameter evaluated, indicate what score 1 corresponds to and what score 7 corresponds to.

L148: It is not correct to start a sentence with a number.

L170: I strongly suggest to present Results and Discussion in separate sections.

L299: replace “studied” with “evaluated”.

Author Response

Dear reviewer,

We really appreciate your kind review for improving the manuscript.

We would like to inform you that we changed total phenolic and flavonoid contents in Table 3 because we realized that the values were miscalculated. We corrected these values and rewrote about them in results and discussion sections. We apologize our mistake.

For one of your major concern, we have considered to separate results and discussions sections, but we didn’t change the structure of the manuscript. Because we think that the revised manuscript has been improved with more discussions, so we would like to confirm the revised manuscript by reviewers again. Separating results and discussions is almost close to rewriting and it will be a big challenge to be perfect within the time limit. Consider about this point, thanks.

Furthermore, some sentences were slightly changed for better readability.

All the modifications indicated in red font. Followings are our detail responses against your comments. We hope the revised manuscript can solve all of your concerns.

Best regards,

Sae-Byuk Lee, Ph.D.

The manuscript submitted by Lee et al. deals with the evaluation of antimutagenic and antioxidant activity of fermented milk supplemented with a powder of Cudrania tricuspidata, a plant containing abundant phytochemicals, such as xanthones and flavonoids.

The study is quite original and the experimental design is clear.

Major criticisms:

  • The English style must be revised.

-> This manuscript was proofread by a native speaker twice, thanks.

  • for each evaluation, the number of samples analyzed and the experimental replicates performed must be clearly indicated.

-> We added experimental replicates in each table and figure, thanks.

  • The C. tricuspidata powder has not been characterized prior to use. The evaluations were made only downstream on fermented milk.

-> We didn’t analyze C. tricuspidata powder but we analyzed the proximate compositions and physicochemical properties of C. tricuspidata fruit. These data are added in Materials and Methods section. Thanks.

  • the discussion needs to be improved. Very often the authors limit themselves to saying only if their results are in agreement or not with what has already been reported in the literature.

-> We added more references to improve discussion sections. Please check revised discussions, thanks.

Minor concerns:

L49: indicate which polyphenols and which flavonoids.

-> We would like to specify which polyphenols and flavonoids in C. tricuspidata can affect to antimutagenic activity. But it has never been studied about the effect of C. tricuspidata on antimutagenic activity before. Because we would guess that there is one more compounds in C. tricuspidata affect to antimutagenic activity, we can’t mention any specific compounds in here to avoid a misconception. Consider about it, thanks.

L50: are these compounds present in equal amount in all parts of the plant (leaves, fruits, roots, ...)?

-> That is a great question. Many studies revealed that leaf and root have higher amounts of phenolics and flavonoids. But we wanted to improve both organoleptic and functional properties using C. tricuspidata fruit. We mentioned this in introduction. Thanks.

L55: must be entered additional information on what is meant by mutagenic activity.

-> We added additional information of mutation and antimutagenicity in separate paragraph. Thanks.

L61: remove “the”.

-> We did it, and we removed “the” in Line 67 as well. Thanks.

L62: “…were obtained from a local from in Sangju…” local company?

-> Sangju is city in Korea. We removed “a local from in” in this sentence. Thanks.

L71: Do you know the relative percentage of Streptococcus thermophilus, Lactobacillus paracasei and L. bulgaricus in the starter mixture?

-> Unfortunately, the company doesn’t provide specific cell numbers of the strains. It would be better if we can use our own lactic acid bacteria. Because the commercial starter is well-developed for fermentation, we didn’t need to use a laboratory LAB because we focused on the effect of C. tricuspidata on fermented milk in this study. Thanks.

L72: It is not correct to start a sentence with a number.

-> We revised it, thanks.

L75: write “C. tricuspidata” in italic.

-> We italicized it, thanks.

L79: paragraph 2.3 must be divided into distinct sections: from L80 to L87, from L88 to L100, from 101 to L110, from L111 to L114.

-> We divided this section as you suggested, thanks.

L92-93: What is the range of the standard curve with gallic acid?

-> We added the range of the standard curve with gallic acid, thanks.

L99: What is the range of the standard curve with quercetin?

-> We added the range of the standard curve with quercetin, thanks.

L115: remove “the”.

-> We removed it, thanks.

L122: What is the range of the standard curve with trolox?

-> We added the range of the standard curve with Trolox, thanks.

L131: “Trolox/1 mL”. Remove “1” and indicate the range of the standard curve with trolox.

-> We revised and added the range of the standard curve with Trolox, thanks.

L141: “Trolox/1 mL”. Remove “1” and indicate the range of the standard curve with trolox.

-> We revised and added the range of the standard curve with Trolox, thanks.

L142: remove “the”.

-> We removed it, thanks.

L148: It is not correct to start a sentence with a number.

-> We removed it, thanks.

L164-165: for each parameter evaluated, indicate what score 1 corresponds to and what score 7 corresponds to.

-> We added a description for each score, thanks.

L170: I strongly suggest to present Results and Discussion in separate sections.

-> We apologize that we didn’t separate Results and Discussion sections. We have considered about reconstructing the manuscript several times, but we believe that a current structure is enough to convey our results to readers. We added another references to improve the discussion of the manuscript. Check them and consider our decision. Thanks for your understanding.

L299: replace “studied” with “evaluated”.

-> We removed it, thanks.

Reviewer 2 Report

Detailed comments and suggestions are given in the annotated manuscript, attached to this report.

The manuscript is interesting and, in general terms, easy to read.

Its originality relies solely on the use of an exotic fruit powder as an antioxidant functional supplement in yougurt. No in vivo studies are provided and the results are sometimes presented under too optimistic a light. English needs some revising by a native speaker. Bacterial taxonomic epithets are outdated and need correcting. Consulting the List of Prokayote Names with standing in Nomenclature is of help in this aspect (https://lpsn.dsmz.de/). Many of the references are to articles that are older than 10 years and therefore need updating. Several parts of the manuscript, especially in the Results and Discussion Section, need clarifying. These, together with several minor comments, are duly marked in the annotated manuscript.

One major flaw in this manuscript is that sensorial testing is internationally regarded as human testing. It necessitates a whole protocol, the signing of informed consent forms by the human subjects and obtaining permission from the ethics commission of the authors' institution. I did not find any proof of such a protocol having been followed in the preparation of the manuscript. The authors should present the permission of the ethics commision and should mention, in the relevant part of the Materials and Methods section, the protocols they followed to ensure ethical compliance.

Author Response

Dear reviewer,

We really appreciate your kind review for improving the manuscript. The annotated manuscript was very helpful to figure out your comments. Now, we transfer all your comment to here to answer your comments one by one.

We have tried to solve your concerns as much as possible.

Firstly, we replaced some old references into newer ones (i.e. ref. 6,7) but most references are key references or only study what we would like to say. So some of old references are still in the manuscript, consider about this point.

Secondly, we realized that some data, such as total phenolic and flavonoid contents in Table 3, were miscalculated. So we corrected these values and rewrote about this in results and discussion sections. We apologize our mistake.

We added discussions with more references and some sentences were slightly changed for better readability. Furthermore, this manuscript was proofread by a professional editing company.

All the modifications indicated in red font. Followings are our detail responses against your comments. We hope the revised manuscript can solve all of your concerns.

Best regards,

Sae-Byuk Lee, Ph.D.

Comments:

The manuscript is interesting and, in general terms, easy to read.

Its originality relies solely on the use of an exotic fruit powder as an antioxidant functional supplement in yougurt. No in vivo studies are provided and the results are sometimes presented under too optimistic a light. English needs some revising by a native speaker. Bacterial taxonomic epithets are outdated and need correcting. Consulting the List of Prokayote Names with standing in Nomenclature is of help in this aspect (https://lpsn.dsmz.de/). Many of the references are to articles that are older than 10 years and therefore need updating. Several parts of the manuscript, especially in the Results and Discussion Section, need clarifying. These, together with several minor comments, are duly marked in the annotated manuscript.

One major flaw in this manuscript is that sensorial testing is internationally regarded as human testing. It necessitates a whole protocol, the signing of informed consent forms by the human subjects and obtaining permission from the ethics commission of the authors' institution. I did not find any proof of such a protocol having been followed in the preparation of the manuscript. The authors should present the permission of the ethics commision and should mention, in the relevant part of the Materials and Methods section, the protocols they followed to ensure ethical compliance.

Line 32: No need for a final s. The word "bacteria" is a plural noun (singular: bacterium).

-> We revised LABs into LAB, thanks.

Line 71: This is no longer the approved name for this taxon. Please correct to Streptococcus salivarius subsp. thermophillus.

-> We think Streptococcus thermophiles is a correct taxon name. We think Streptococcus salivarius subsp. Thermophillus what you mentioned is an old taxon. Check it please.

Line 71: The Lactobacillus paracasei species has two subspecies. Which was used here, Lb. paracasei subsp. paracasei or Lb. paracasei subsp. tolerans?

-> The company doesn’t provide subspecies of Lactobacillus paracasei. Because it can be a sensitive issue, we can only use officially provide strain name. You can also confirm this article (see below link) that the company provided the strain name, L. paracasei LPC5. https://onlinelibrary.wiley.com/doi/epdf/10.1111/jgh.14362

Line 71: This is no longer the approved name for this taxon. Please correct to Lactobacillus delbrueckii subsp. bulgaricus.

-> We agree Lactobacillus bulgaricus is an old taxon name. So we replaced this into the latest taxon name as you suggested, thanks.

Line 74: The formulation of this fragment of the sentence is not clear. I suggest replacing with "the fermentation substrate for yogurt production".

-> ‘a mixture of fermented milk’ was replaced into ‘the fermentation substrate for yogurt production’, thanks.

Line 75: The word "sterilized" should not be used here. This temperature-time combination would not destroy bacterial spores eventually present in the milk. Therefore, it cannot be touted a sterilization process.

-> ‘sterilized’ was replaced into ‘heat-treated’. Thanks.

Line 75: ‘C. tricuspidata’ Italics missing.

-> We italicized it, thanks for pointing it.

Line 79: English needs revision in this title. Also, it should specify which viable cells the authors are counting, i.e., the LAB. Therefore, I would propose to use "Assessment of physicochemical properties and LAB viable counts in the fermented milk" as a title.

-> We revised a title as you suggested, thanks.

Line 83: Please replace with "was determined by titration".

-> ‘was titrated’ was replaced into ‘was determined by titration’

Line 98: ‘were’ The verb here has to agree with 500 ul and not with NaOH.

-> We think ‘was’ is correct in this sentence. Because the quantity such as μL takes a singular verb form. See below link. Thanks.

https://www.aje.com/arc/editing-tip-singular-and-plural-verbs-measured-quantities/

Line 115: I do not completely agree with the use of "measurement" in this context. This is more of an assessment or evaluation. Furthermore, the initial "The" should be omitted.

-> We revised them according to your suggestion, thanks.

Line 116: Delete ‘the’.

-> We deleted it, thanks.

Line 142: Here, again, a word of caution against applying the word "measurement" to experimental processes that are, at best, semi-quantitative. Use instead evaluation, assessment, etc.

Delete the initial "the".

-> We revised it, thanks.

Line 147: Presently, this taxon is again regarded as a species, so the name should be written as Salmonella typhimurium.

-> We revised them according to your suggestion, thanks.

Line 159: In accordance with the present international standards, sensory evaluation is regarded as human experimentation. Therefore, the authors should present the approval of their experiment by the ethical commission of their institution - or omit this part altogether - for the manuscript to be accepted for publication.

-> We provided ethical information for sensory evaluation, thanks.

Line 180: How do these compare with C. tricuspidata in terms of phenolic content? Often, phenolic compounds are also antibacterial and detrimental for LAB growth. It is, therefore, important to compare the authors' results with others that have a similar phenolic content. This sentence needs rewriting and, if necessary, using different examples, with the aim of making clear to the readers that the authors are comparing their data with other studies with materials that have a similar content of phenolics.

-> This is a great point. As you mentioned, phenolic compounds such as tannic acid or catechine have been revealed to have antimicrobial activity. But we don’t think that all the phenolics negatively affect to lactic acid bacteria. Most of studies about antimicrobial activity of phenolics are related to wine. Many studies have reported that phenolic compounds are beneficial for LAB. In fact, The International Scientific Association for Probiotics and Prebiotics (ISAPP) officially defined that prebiotics include polyphenols and phytochemicals (see below link). Furthermore, we agree with different examples are needed in the manuscript. So, we replaced references and rewrote this paragraph. Please check revised sentences, thanks.

https://www.nature.com/articles/nrgastro.2017.75.pdf?origin=ppub

Line 186: The authors should caution the readers that these are in vitro results and that in vivo results might differ. The final aim of ingesting flavonoid-enriched foods is to obtain a desirable flavonoid content in the blood plasma. The casein matrix in dairy products does hinder the absorption of phenolics, so the authors should give a word of caution here and should not sound overly optimistic about the promising, albeit limited, results they obtained.

-> Thanks for pointing this. We agree with your opinion. This research was conducted in vitro so we need to confirm functionality of fermented milk by further in vivo study. Therefore, we removed this sentence in this paragraph and mentioned the necessity of in vivo study in the conclusion section. Thanks.

Line 194: Revise English ‘like as viable cell count of each sample’

-> We revised this sentence, thanks.

Line 194: Comments on this observation? Would the authors expect that the LAB species in their starter would consume galactose? Or can they offer another explanation?

-> We added another explanation about lactose and galactose in this sentence. Thanks.

Line 204 and Table 2: Revise ‘viable LAB cell count’

-> We revised it as you suggested, thanks.

Line 227: Revise ‘more yellow’

-> We added ‘more’ here. Thanks.

Line 250: Again, a word of caution here. The authors should stress that these are in vitro results and that they do not always translate directly into evidence of health benefits or even into higher plasma levels of antioxidants when applied in vivo. Furthermore, antioxidants are one of the cases in which more is not always better. Indeed, at sufficiently high concentrations, phenolics such as those present in the plant powder under study, can act as pro-oxidants and have, therefore, detrimental effects on the users' health.

-> We revised this sentence. Please check it, thanks.

Line 256, 258, and 275: See previous comment on this taxonomic name. Salmonella typhimurium

-> We revised them according to your suggestion, thanks.

Line 293: This aspect should be better debated in terms of how much improvement in each functional parameter the authors evaluated the addition or 1% powder, instead of 2 or 3% still provides, as this is an important consideration for the future applicability of the yogurt formula under assessment.

-> We think 1% addition of C. tricuspidata powder have also great functionality even this is in vitro test. So we would think that providing certain values (i.e. 1.5-3 times higher antioxidant activity) for 1% addition condition can help future readers’ understanding. Thanks.

Line 300: This statement is too vague. In scientific writing, using such umbrella terms when stating your conclusions may be misleading for the readers. Pleas modify the sentence, making it less vague and more precise.

-> We specified ‘bioactive compounds’ in this sentence. Thanks.

Line 309: One could argue that the functionality provided by these lower concentrations of the fruit powder are not excellent, as the ones provided by higher quantities are even better. Therefore, I would again caution once more against being overly optimistic when reporting your conclusions. Therefore, I would propose to change the sentence for: "while still providing good functional properties".

-> We revised this sentence according to your suggestion, thanks.

Reviewer 3 Report

The work presented for review concerns issues that are currently relevant and important in terms of the demands of modern consumers, who are increasingly interested in products that are beneficial to the functioning of the body. The authors evaluated how C. tricuspidata improves the functional properties of fermented milk and determined the optimal amount of C. tricuspidata that would be organoleptically acceptable to potential consumers. The aim of the study is clearly defined, but the description of the material and research methods is inadequate.  I suggest including a photo of the fruits of C. tricuspidata, as they are not common. The authors should also specify what standards and calibration method were used to determine free sugar and organic acids. The discussion should be expanded to attempt to explain the results of the research.

Author Response

Dear reviewer,

We really appreciate your kind review for improving the manuscript.

We would like to inform you that we changed total phenolic and flavonoid contents in Table 3 because we realized that the values were miscalculated. We corrected these values and rewrote about them in results and discussion sections. We apologize our mistake.

All the modifications indicated in red font. Followings are our detail responses against your comments. We hope the revised manuscript can solve all of your concerns.

Best regards,

Sae-Byuk Lee, Ph.D.

The work presented for review concerns issues that are currently relevant and important in terms of the demands of modern consumers, who are increasingly interested in products that are beneficial to the functioning of the body. The authors evaluated how C. tricuspidata improves the functional properties of fermented milk and determined the optimal amount of C. tricuspidata that would be organoleptically acceptable to potential consumers. The aim of the study is clearly defined, but the description of the material and research methods is inadequate.  I suggest including a photo of the fruits of C. tricuspidata, as they are not common. The authors should also specify what standards and calibration method were used to determine free sugar and organic acids. The discussion should be expanded to attempt to explain the results of the research.

  • I appreciate your kind comments. We added a photo of tricuspidata fruit, detail information about standards of free sugar and organic acid measurements. Finally, we added more discussion to support our results. Thank you so much.

Round 2

Reviewer 1 Report

All criticisms raised during the review process have been adequately addressed.